# Flame Retardant Submicron Particles via Surfactant-Free RAFT Emulsion Polymerization of Styrene Derivatives Containing Phosphorous

**DOI:** 10.3390/polym12061244

**Published:** 2020-05-29

**Authors:** Taeyoon Kim, Joo-Hyun Song, Jong-Ho Back, Bongkuk Seo, Choong-Sun Lim, Hyun-Jong Paik, Wonjoo Lee

**Affiliations:** 1Center for Advanced Specialty Chemicals, Korea Research Institute of Chemical Technology, Ulsan 44412, Korea; xodbs17@krict.re.kr (T.K.); joohyun@krict.re.kr (J.-H.S.); beak1231@krict.re.kr (J.-H.B.); bksea@krict.re.kr (B.S.); chsunlim@krict.re.kr (C.-S.L.); 2Department of Polymer Science and Engineering, Pusan National University, Busan 46241, Korea

**Keywords:** phosphorus, flame retardant, RAFT, emulsion polymerization, surfactant-free

## Abstract

The reversible addition–fragmentation chain transfer (RAFT) emulsion polymerization of diethyl-(4-vinylbenzyl) phosphate (DEVBP) was performed using PEG-TTC as a macro RAFT agent. PEG-TTC (M_W_ 2000, 4000) was synthesized by the esterification of poly (ethylene glycol) methyl ether with a carboxylic-terminated RAFT agent, composed a hydrophilic poly (ethylene glycol) (PEG) block and a hydrophobic dodecyl chain. The RAFT emulsion polymerization of DEVBP was well–controlled with a narrow molecular size distribution. Dynamic light scattering and confocal laser scanning microscopy were used to examine the PEG-b-PDVBP submicron particles, and the length of the PEG chain (hydrophilic block) was found to affect the particle size distribution and molecular weight distribution. The submicron particle size increased with increasing degree of polymerization (35, 65, and 130), and precipitation was observed at a high degree of polymerization (DP) using low molecular weight PEG-TTC (DP 130, A3). The flame retardant properties of the PEG-b-PDVBP were evaluated by thermogravimetric analysis (TGA) and micro cone calorimeter (MCC). In the combustion process, the residue of PEG-b-PDEVBP were above 500 °C was observed (A1 ~ B3, 27 ~ 38%), and flame retardant effect of PEG-b-PDEVBP submicron particles/PVA composite were confirmed by increasing range of temperature and decreasing total heat release with increasing contents of PEG-b-PDEVBP. The PEG-b-PDEVBP submicron particles can provide flame retardant properties to aqueous, dispersion and emulsion formed organic/polymer products.

## 1. Introduction

Phosphorus-based polymers have many applications, such as organic-metal complex [1], biominerals [2], proton exchange membrane materials [3]. In particular, phosphorous organic compounds and polymers have flame retardant properties to prevent fire, and minimize damage from fire, through the physical incorporation of fire retardant additives [4]. Fire retardant polymers generally contain elements, such as halogens, phosphorus, silicon, and boron. Although halogen-based flame retardants show good performance, they generate toxic and corrosive gases, such as Cl_2_, HCl, Br_2_, HBr, and HF. Phosphorus-based flame retardants have been evaluated as alternative halogen flame retardants [5]. Phosphorinated halogen-free retardant coatings can lead to a char or a protective coating that prevents oxygen transport toward the burning area resulting in the extinguishing of a fire [5]. On the other hand, the physical incorporation of low molecular phosphorus compounds has some problems, such as the requirement of additives and lack of compatibility with organic, polymeric materials [6]. Therefore, flame retardant additives are needed to form polymers, copolymers, and modify the chemical structure.

Over the past few decades, reversible deactivation radical polymerization (RDRP) methods, such as atom transfer polymerization (ATRP) [7], stable free radical polymerization (SFRP) [8], and reversible addition–fragmentation chain transfer (RAFT), have been developed [9]. These techniques can allow control of the molecular weight with narrow polydispersity, as well as various structure polymers, and polymer chain-like blocks, grafts, stars, and gradient copolymers. In particular, RAFT polymerization is a simple process because of its relatively mild reaction conditions, lack of metal catalysts, and a wide range of monomers [10]. Heterogeneous RAFT polymerization has advantages, both industrially and environmentally, such emulsion, miniemulsion, self-assembly, and surfactant-free polymerization [11]. Conventional emulsion/dispersion polymerization uses a surfactant to stabilize polymeric particles, and RAFT emulsion polymerization has also used surfactant [12]. The final polymer product, however, has adverse effects on the properties of the products, due to surfactant migration through the polymeric material, leading to degradation of their adhesive and mechanical properties. In addition, they also harm the environment [12]. RAFT polymerization can provide surfactant-free emulsion polymerization because of its living character [12]. Recently, surfactant-free RAFT emulsion polymerization was shown that the particle size and shape size can be controlled, and it was widely used in the biomedical field [13,14].

Matyjaszewski et al. we dimethyl(1-ethoxycarbonyl)vinyl phosphate (C–O–PO– (OCH_3_)_2_) [15], dimethyl (methacryloyloxymethyl) phosphonate (C–PO– (OR)_2_, PO_3_) [16], with a narrow molecular weight distribution. Dong et al. reported the ATRP of diethyl-(4-vinylbenzyl) phosphonate using polypropylene macroinitiator (PO_3_), and NMP of diethyl-(4-vinylbenzyl) phosphonate (PO_3_) was reported (PO_3_) [5,17]. Poly(methacryloyloxy methyl phosphonic acid) were directly synthesized by RAFT polymerization using methacryloyloxymethyl phosphonic acid, and synthesized by RAFT polymerization using methacryloyloxymethyl dimethylphosphonate followed by conversion to poly(methacryloxymethylphosphonic acid). Poly(methacryloxymethylphosphonic acid) was used for emulsion RAFT polymerization as the hydrophilic macro-CTA [18,19].

In this study, PEG-b-P(diethyl-(4-vinylbenzyl) phosphate (DEVBP))-based submicron particles were prepared by the RAFT emulsion polymerization of DEVBP, which was used as a hydrophobic monomer, using a poly (ethylene glycol)-based chain transfer agents. DEVBP was PO_4_ based styrene derivatives, which has been so far used in the conversional radical polymerization [20,21], and which monomer was first used for RAFT and emulsion RAFT polymerization as the hydrophobic macro-CTA. Furthermore, PO_4_-functionalized compounds/polymers show a good performance for flame retardant compared with –PO_1_, –PO_2_, –PO_3_ [22,23]. Characterization of PEG-b-PDEVBPs were confirmed by ^1^H and ^31^P NMR, gel permeation chromatography (GPC) and PEG-b-PDEVBP submicron particles were observed by dynamic light scattering (DLS), confocal laser scanning microscope (CLSM). Flame-retardant properties of PEG-b-PDEVBP were performed by TGA, micro cone calorimeter (MCC). The synthesis of PEG-b-PDEVBP is illustrated below (Figure 1).

## 2. Materials and Methods 

### 2.1. Materials

4-vinylbenzyl chloride was purchased from Tokyo Chemical Industry Co., Ltd. (Tokyo, Japan) and dried by distillation prior to use. Diehtyl chlorophosphate was purchased from Tokyo Chemical Industry Co., Ltd. and used without further purification. 4,4′-azobis-4-cyanopentanoic acid (ACPA) was purchased from Wako Pure Chemical Co., Ltd. (Osaka, Japan) and dried by vacuum oven at room temperature before to use. Methoxy poly(ethylene glycol) (mPEG), 4-Dimethylaminopyridine (DMAP) and N,N’-dicyclohexylcarbodiimide (DCC) were purchased from Alfa Aesar Co. (Waltham, MA, USA) Dichloromethane (DCM), dimethyl sulfoxide (DMSO) were purchased from Daejung Chemical & Metal Co. (Siheung, Korea) and purified by distillation over CaH_2_. 4-Cyano-4-[(dodecylsulfanylthiocarbonyl)sulfanyl]pentanoic acid was synthesized by the reaction procedure according to the similar method reported elsewhere [24]. All other chemicals were used without further purification. 

### 2.2. Characterization

^1^H NMR and ^31^P NMR spectra were recorded in CDCl_3_ on Bruker 300 Mhz spectrometer. FT-IR spectra were obtained at a resolution of 4 cm^−1^ with ThermoFisher Nicolet 6700 spectrometer (Waltham, MA, USA) in the wavenumber rage of 4000 ~ 400 cm^−1^. The powder samples were incorporated into KBr pellets and IR measurements were performed. Thermal degradation was examined using a TA Q500 thermogravimetric analyzer (TGA) with heating rate of 10 °C/min under nitrogen atmosphere and samples were dried by the lyophilization. The heat resistance was studied with pyrolysis combustion flow calorimetry (PCFC, Fire Testing Technology Limited, West Sussex, UK) with the ASTM D7309 method. The samples were prepared by the mixing 10 wt% PVA solution with PEG-b-PDEVBP emulsion followed by the lyophilization of the mixture and then loaded in the instrument is heated at a rate of 1 C/s from 100 to 750 °C to obtain heat release rate (HRR) and total heat release (THR). Molecular weight and polydispersity were determined by (GPC), conducted with a Agilent 1260 isopump and Agilent 1260 differential refractometer using Agilent columns (Santa Clara, CA, USA) (2× PLgel 5 μm MIXED-D, 7.5 × 300 mm) in THF as an eluent at 40 °C and at a flow rate of 1 mL/min. Linear polystyrene standards were used for calibration. Particles size measurement was performed on Malvern Nano-ZS90 (Malvern, UK). Multi-photon confocal microscopy images were acquired using a ZEISS LSM 780 (Oberkochen, Germany) configuration 16 NLO microscope with Nile red as fluorescent agent. To obtain the submicron particles stained with Nile red, DEVBP containing 0.1wt% Nile red was used for RAFT emulsion polymerization. 

### 2.3. Synthesis of 4-Vinylbenzyl Alcohol (VBA)

4-vinylbenzyl alcohol was synthesized as described before [25]. 4-vinylbenyl chloride (30.5 g, 0.2 mol), potassium acetate (25 g, 0.25 mol), 0.05 g of t-butylcatechol were add in 3 neck round flask and dissolved in DMSO 200mL. The reaction mixture was stirred 20 h at 40 °C under nitrogen. After reaction, 400 mL ether was added in the reaction mixture and then washed with distilled water and dried over Na_2_SO_4_. Colorless liqud 4-vinylbenzyl acetate was obtained. The obtained 4-vinylbenzyl acetate was hydrolyzed with a mixture of ethanol and potassium for 10 h at 50 °C under nitrogen. The product was wahsed with ether and water followed by dried over anhydrous Na_2_SO_4_.

### 2.4. Synthesis of Diethyl 4-Vinylphenyl Phosphate (DEVBP) 

Diethyl 4-vinylphenyl phosphate was synthesized as discribed previously [20,25]. VBA and pyridine added a three necked round bottom flask, supplied with a dropping funnel and dissoved in dichlromethane, and then the reactin mixture cooled at 0 °C in an ice bath. Diethyl chlorophosphate dissolved in dicrhloromethane was dropwised in the reaction mixture and was sttired for 24 h at ambient temperature. After the reaction, pyridine·HCl was filtered follwed by filtrate was wished with distilled water 3 times. The organic layer dried with Na_2_SO_4_. Color less liquid product was obtained after dirchloromethane was evaporated.

### 2.5. Synthesis of PEG-TTC (macro-CTA)

PEG-TTC was synthesized by the reaction procedure according to the similar method reported elsewhere [26]. Typical procedure for the synthesis of PEG-TTC, mPEG and DMAP dissoved in 50 mL dichlromethane 250 mL in 3 neck flask. DCC in 40 mL of dichloromethane was dropwised in the reaction mixture for 1h under nitrogen atmosphere and the reaction mixture was sttirred 120 h. After reaction, pricipitated dicyclohexylurea was removed by filteration and the filtrate was pricipitated in cold ehtyl ethyl ether. The pricipitated product was dried in vacuum oven at ambient temperature for 24 h and sligtly yellow powder was obtained (See Figure 3). 

### 2.6. RAFT Emulsion Polymerization of Diethyl 4-Vinylphenyl Phosphate (DEVBP) 

PEG-TTC 4k (0.27 g, 0.060 mmol), ACPA (0.012 g, 0.042 mmol) were added to a 100 mL schlenk flask and dissolved in water. DEVBP (1.06 g, 3.94 mmol) was added to the reaction mixture and dispersed. An aqueous stock solution of neutralized ACPA by NaHCO_3_ (3.5 molar equivalents with respect to ACPA) was prepared and added. Freeze-pump-thaw cycles were performed 3 times toremove oxygen. After the flask was backfilled with nitrogen, the polymerization was carried out at 70 °C for 6 h. Adequate samples were periodically withdrawn from the reactor analyzed by by ^1^H NMR to evaluate polymerization kinetics, conversion was calculated by the comparsion vinyl peak with methylene peaks of DEVBP.

## 3. Results and Discussion

The submicron particle size and distribution were characterized, and the flame retardant properties were investigated. PEG-TTC was synthesized by reacting PEG with CPD-TTC and RAFT emulsion polymerization of DEVBP with PEG-TTC (Figure 1). DEVBP was synthesized using a similar method to that reported elsewhere [20,25], and its structure was confirmed by ^1^H and ^31^P nuclear magnetic resonance (NMR) spectra (Figure 2). In ^1^H NMR spectrum of 4-vinylbenzyl acetate (Figure 2), ω-methylene proton (–CH_2_–Cl) was shifted 4.79 to 5.08 ppm with the appearance of methyl (–CH_3_–) at 2.07 ppm by the reacting 4-vinylbenyl chloride with potassium acetate. The VBA was synthesized by depreotection reaction, and confirmed by the shifting methylene proton (–CH_2_–C–O) of 4-vinylbenzyl acetate at 5.08 ppm to 4.48 ppm. Finally, DEVBP, styrene derivatives containing phosphorus, were obtained by the esterification of VBA with diethyl chlorophosphate. In ^1^H NMR and ^31^P NMR, DEVBP were identified by the appearance of methylene and methyl proton at 4.03, 1.23 ppm corresponding to diethyl chlorophosphate (–P–O–CH_2_–CH_3_) and the shifting phosphorus peak from −0.98 to 4.25 ppm. 

A PEG-based macro RAFT agent have used for the emulsion RAFT polymerization of styrene [27], methyl methacrylate [28], butyl acrylate [26,28]. In previously, polymerization of styrene was very slow and low conversion in the presence of a PEG based dithiobenzoate macro RAFT agent [26,27]. On the other hand, a trithiocarbonate -based macro-RAFT agent, which was synthesized by reacting mPEG with 2-(dodecylthiocarbonothioylthio)-2-methylpropanoic acid), was used successfully as both a stabilizer and a reversible chain transfer agent in the ab initio batch emulsion polymerization of styrene, n-butyl acrylate, and methyl methacrylate, and the stable latexes were obtained [26,27,28]. The synthesized PEG-TTC (2k, 4k) were confirmed by the shift in the methylene group (–CH_2_–C–) of TTC at 2.75 to 4.25 ppm in the ^1^H NMR spectra (Figure 3a), as well as by the disappearance of a hydroxyl peak at 3000 ~ 3500 cm^−1^, and the appearance of a carbonyl peak at 1750 cm^−1^ in the FTIR spectra (Figure 3b). In gel permeation chromatography (GPC), the molecular weight of PEG-TTC was increased slightly after esterification with TTC (Figure 3c). 

Functionality of TTC was above 92%, as determined by comparing the integration of methyl protons (CH_3_–CH_2_–) of TTC at 0.85 ppm with methylene protons (–CH_2_–CH_2_–O–) of mPEG at 4.25 ppm. RAFT emulsion polymerization of DEVBP was conducted using PEG-TTC as an amphiphilic chain transfer agent and 4,4’-azobis(4-cyanopentanoic acid) (ACPA) as an initiator at 70 °C (Table 1). The copolymerization of diethyl-(4-vinylbenzyl) phosphate (DEVBP) (–C–O–PO–(OC_2_H_5_)_2_) was performed by free radical polymerization [21,25] and RAFT polymerization. In the current study, a phosphorus- containing styrene-derivative monomer (DEVBP) was first synthesized by RAFT emulsion polymerization and used as the hydrophobic moiety. The PEG-based trithiocarbonate macro RAFT agent consisted of poly (ethylene glycol) (hydrophilic moiety) and dodecyl aliphatic hydrocarbon chain (hydrophobic moiety). In the RAFT emulsion polymerization of DEVBP, PDVBP with a controlled molecular weight and narrow molecular weight distribution was synthesized within 6 h (83% <= conversion) (Figure 4). The molecular weights of PEG-b-PDVBP increased linearly with increasing conversion, which is the supporting evidence for controlled polymerization (Figure 4). No induction period for the homopolymerization for DEVBP was observed (Figure 4). However, it was shown that the tailing to low molecular weight, and the molecular weight of all polymers (Figure 5, A1–B3) was smaller in GPC than the actual molecular weight calculated by ^1^H NMR spectroscopy (Table 1). In a previous paper, the polymerization of styrene, using PEG-TTC tailing, was not observed at high monomer conversion, and symmetrical curve was shown in GPC [26]. The hydrodynamic volume of PEG-b-PDEVBP series, a polymer of DEVBP derived from styrene, were very small compared with polystyrene standard. A strong intermolecular interaction occurred between the phosphate moieties of PEG-b-PDEVBP, leading to tailing toward low molecular weight (Figure 4). The polymerization behavior of DEVBP differed according to the molecular weight of PEG-TTC (2k, 4k). In cases of using PEG-TTC 2k, PDVBP was obtained with a relatively narrow molar mass distribution (M_n_/M_w_: 1.07 ~ 1.08) than PEG-TTC4k (M_n_/M_w_: 1.15 ~ 1.20) (Table 1, Figure 5). The vinyl peaks at 5.7, 6.7 ppm, and the polymer backbone at 1 ~ 2 ppm disappeared and appeared, respectively, in the ^1^H NMR spectra (Figure 6 and Figure 7) after polymerization. In addition, phosphorus peaks were also observed in the ^31^P NMR spectra (Figure 6 and Figure 7). The PEG-b-PDVBP particles were obtained by emulsion RAFT polymerization (Figure 8 and Figure 9). Particles size increased with increasing molecular weight in emulsion RAFT polymerization previously [29], the PDVBP particle size also increased with increasing target molecular weight (z- average size 216 ~ 1056 nm). On the other hand, the particle size distribution increased by the degree of polymerization (DP). The particle size of A2 (DP 65) was larger, and the particle distribution was much broader than A1 (DP 32). Furthermore, the precipitation of particles was observed. Short chain PEG 2K could not stabilize PDVBP with a high DP (Figure 6). In contrast, in a series of experiments (B1 ~ B3), PEG –TTC 4k was used as an amphiphilic RAFT agent. All of the PDVBP particles were obtained with a stable and narrow size distribution (194 ~ 576 nm). The PEG-TTC 4k-based particles were smaller than the PEG-TTC 2K-based particles, PEG-TTC 2K-based particles aggregated because of their short hydrophilic chain. Phosphorus compounds can provide excellent thermal stability and show enhanced char formation during the combustion process. 

The flame retardant properties were analyzed by thermogravimetric analysis (TGA) with a heating rate of 10 °C/min under air. The residual weight at 500 °C, according to the weight percentage of phosphorus, was confirmed. The residual weight increased with increasing phosphorus weight percentage in PDVBP (Figure 10). PEG-TTC 2k, 4k and polystyrene mainly degraded at 380 °C and were degraded entirely at 400 °C. The residue was 0% above 400 °C. Different thermal degradation behaviors were observed in PDVBP. The thermal degradation of PDVBP (P contents: 9 ~ 10.2%) started at 250 to 300 °C; the residue weight was above 60%. The initial degradation of PDVBP was contributed by the P-O-C structure functional group, P-O-C group of -PO4 is unstable than -PO_1-3_. Previously, -PO_4_ group was more effective for phosphorus release and forming char [22]. The second thermal degradation occurred at 400 ~ 430 to 470 °C, which was contributed by aromatic and PEG; the residue weights were 27.1 ~ 38.1% (A1 ~ B3). Thermal degradation of aromatic group and PEG-TTC were occurred at high temperature (above 450 °C) compared with polystyrene (410 °C) and PEG-TTC (380 °C). It was evidenced to form char by phosphorus contents and PEG-b-PDEVBP submicron particles can provide flame retardant properties. After 500 °C, the residue weights were constant. The residue weight was more than the P contents. Therefore, phosphorus can provide flame retardant properties. The thermal degradation behavior of PEG2k-b-PDVBP (A1 ~ A3) differed slightly according to the P contents (9.0 ~ 10.7). The final residue weight at 500 °C (27.1 ~ 37.8) and second thermal degradation temperature (410 ~ 430 °C) increased with increasing P content. Although the thermal degradation behavior of PEG2k-b-PDVBP (B1 ~ B3) differed according to the P content (7.6 ~ 10.2), the residue weights were similar (24.5 ~ 38.1%). Poly(vinyl alcohol) (PVA) is water soluble polymer and widely used for adhesive. To confirmed flame retardant properties of PEG-b-PDEVBP (B2) as an additive, TGA and micro combustion calorimetry were used as an instrument for assessing the flame retardancy of PEG-b-PDEVBP/PVA composite (Figure 11). PEG-b-PDEVBP submicron particles were added in PVA based on phosphorus contents (0, 3%, 5%), PVA was dissolved in water and added PEG-b-PDEVBP submicron particles. Thermal degradation of PVA were carried out in multi-step, first main thermal degradation was shown at 323 °C and then the thermal degradation was complete at 550 °C (Figure 11a). In case of PVA/PEG-b-PDEVBP, initial thermal degradation temperature was decreased with PEG-b-PDEVBP contents but final thermal degradation temperature was increased (Figure 11a). There were a big difference of thermal degradation behavior between PVA and PVA/PEG-b-PDEVBP, thermal degradation behavior of PVA/PEG-b-PDEVBP according to phosphorous contents were a little. The inflection point of thermal initial decomposition temperature were slightly decreased with increasing phosphorous contents (1% → 3%, 272 °C → 266 °C). In the results of micro cone calorimeter (MCC) (Figure 11b), samples were heated at a rate of 1 °C/min from 100 to 750 °C. As the heat release rate (HRR) value, several overlapping peaks around the maximum value of 356 °C were observed. Overall HRR peaks of PVA/PEG-b-PDEVBP were distributed over a wider temperature range than PVA’s one in Figure 11b. The initial temperature and value of HRR decreased, and the final temperature of HRR peaks increased by the increasing PEG-b-PDEVBP contents. It is indicated that phosphorus severs to delay heat dissipation and total heat release was also decreased by increasing PEG-b-PDEVBP. In results of MCC, THR and heat release time were significantly increased with phosphorous contents. Therefore, it was founded that PEG-b-PDEVBP could provide to give flame retardant property.

## 4. Conclusions

The RAFT emulsion polymerization of DEVBP was conducted using a PEG-based amphiphilic RAFT agent. The well-defined PEG-b-PDVBP was obtained, and the behavior of RAFT emulsion polymerization for DEVBP was examined. PEG2k-PDVBP (M_n_/M_w_: 1.07 ~ 1.08) showed a narrower molecular weight distribution than PEG4k-DEVBP (M_n_/M_w_: 1.15 ~ 1.35). The z-average sizes of PEG2k-b-DEVBP were 216 ~ 1056 nm with a broad size distribution. A relatively short PEG2K chain was insufficient to stabilize PDEVBP block (above 65), on the other hand the submicron particles of PEG4k-b-DEVBP were obtained with relatively low particle size distribution than submicron particles of PEG2K-b-DEVBP. TGA and MCC were analyzed to examine the flame resident property of PEG-b-PDEVBP and PVA/PEG-b-PDEVBP as an additive. The residue weights of PEG-b-PDEVBP at 500 °C were 27.1 ~ 38.1% and thermal degradation temperature were also shift to high temperature in TGA measurements, which confirms that phosphorus contents provide flame retardant properties. The PEG4k-b-PDEVBP submicron particles were added in PVA and then flame retardant effect was confirmed by TGA and MCC analysis, thermal degradation temperature of PVA/PEG-b-PDEVBP increased. Moreover, HRR values were distributed over a wider temperature range and THR values decreased than PVA. Therefore, PEG-b-PDEVBP submicron particles can be assessed further as an additive for flame retardant organic and polymeric materials.

## Figures and Tables

**Figure 1 polymers-12-01244-f001:**
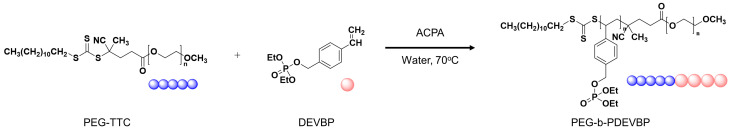
Synthesis of PEG-b-PDVBP.

**Figure 2 polymers-12-01244-f002:**
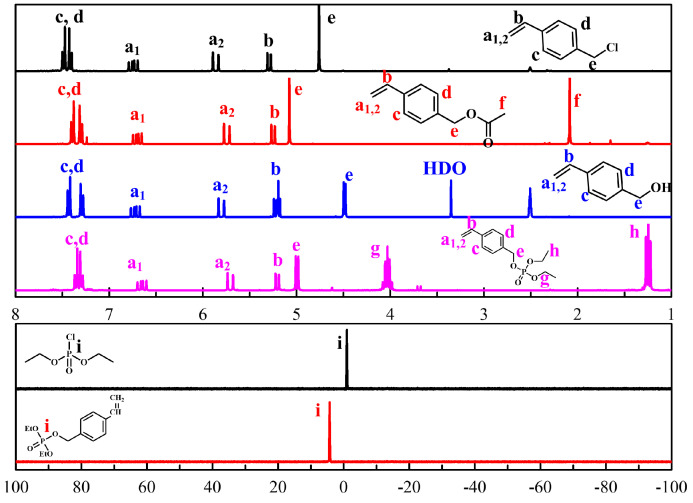
^1^H (4-vinylbenyl chloride, 4-vinylbenzyl acetate, 4-vinylbenzyl alcohol, diethyl 4-vinylphenyl phosphate) and 31P NMR spectra (diethyl chlorophosphate, diethyl 4-vinylphenyl phosphate).

**Figure 3 polymers-12-01244-f003:**
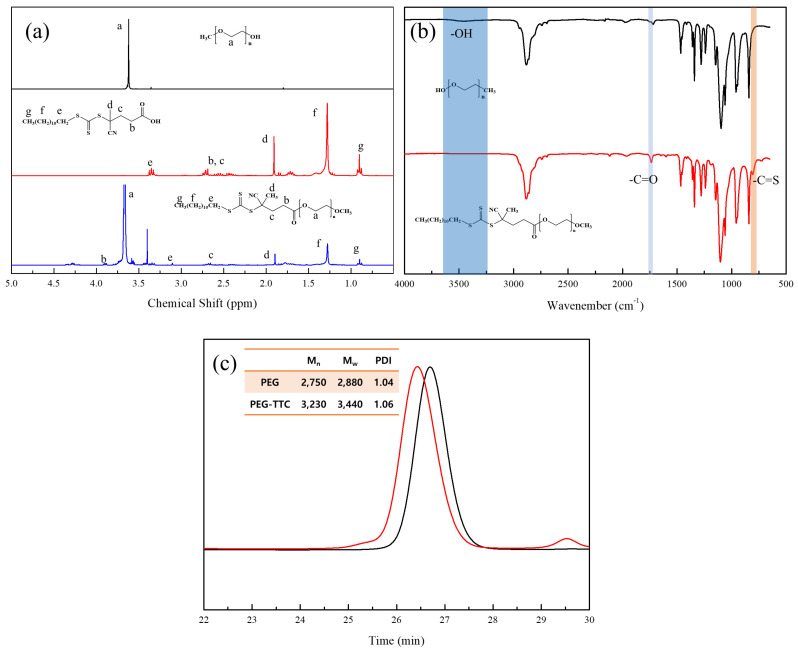
Macro RAFT agent (PEG-TTC) of (**a**) ^1^H NMR spectra, (**b**) FTIR spectra, (**c**) GPC measurements.

**Figure 4 polymers-12-01244-f004:**
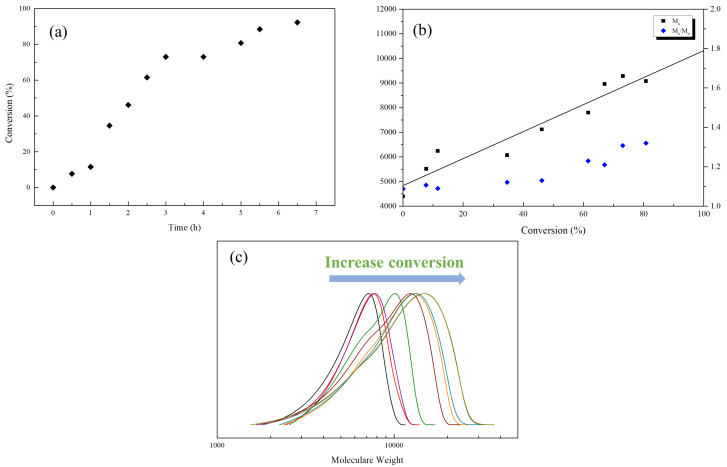
RAFT emulsion polymerization of DEVBP (B2) using poly (ethylene oxide)-based trithiocarbonate at 70 °C: (**a**) monomer conversion determined by ^1^H NMR vs time, number average molar weight; (**b**) M_n_, and polydispersity index, M_w_/M_n_, determined by GPC (polystyrene calibration) vs conversion; the straight line corresponds to the theoretical M_n_ vs conversion (**c**) evolution of GPC with conversion.

**Figure 5 polymers-12-01244-f005:**
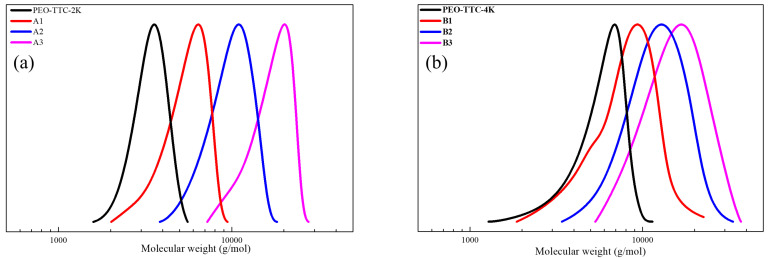
GPC trace of (**a**) PEG2k-b-PDVBP, (**b**) PEG4k-b-PDVBP.

**Figure 6 polymers-12-01244-f006:**
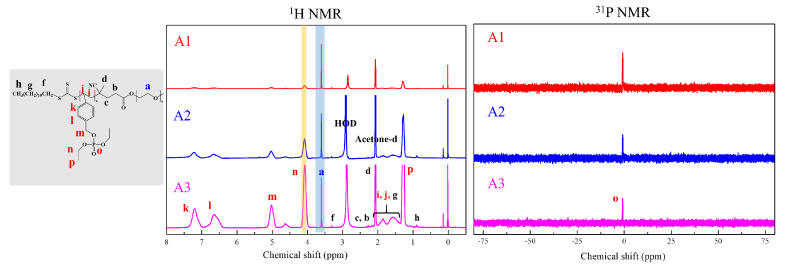
^1^H and ^31^P NMR of PEG2k-b-PDVBP.

**Figure 7 polymers-12-01244-f007:**
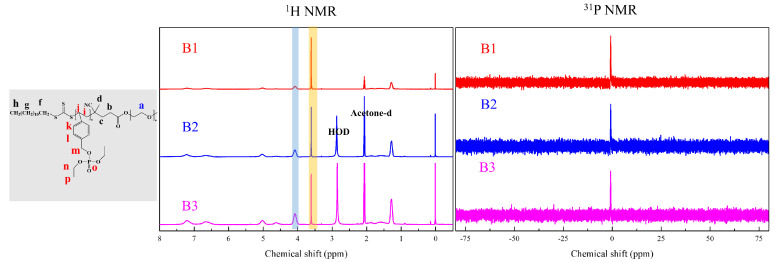
^1^H and ^31^P NMR of PEG4k-b-PDVBP.

**Figure 8 polymers-12-01244-f008:**
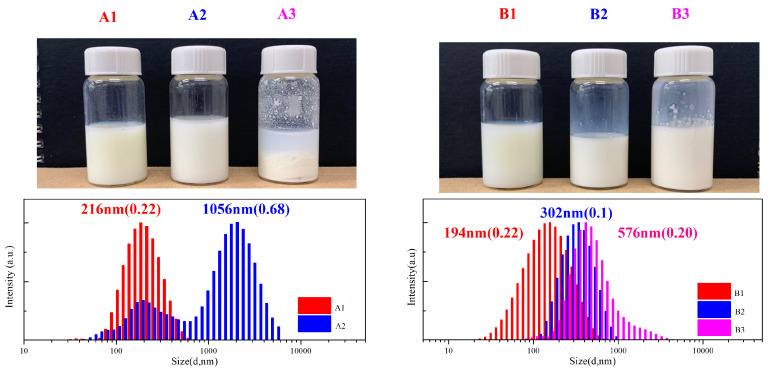
Emulsion polymerization and DLS measurements of PEG-b-PDVBP.

**Figure 9 polymers-12-01244-f009:**
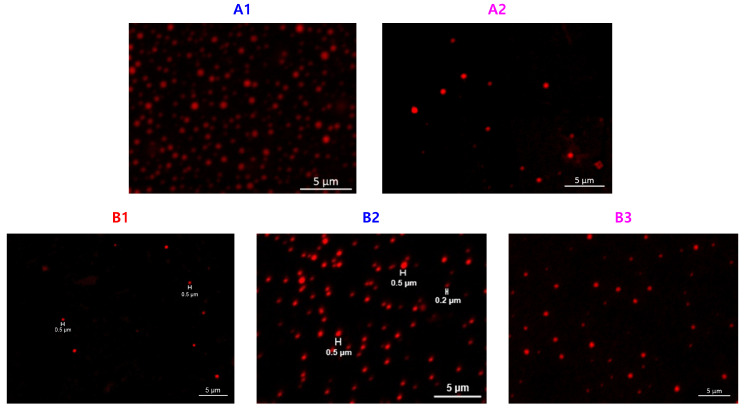
Confocal images of PEG-b-PDVBP submicron particles.

**Figure 10 polymers-12-01244-f010:**
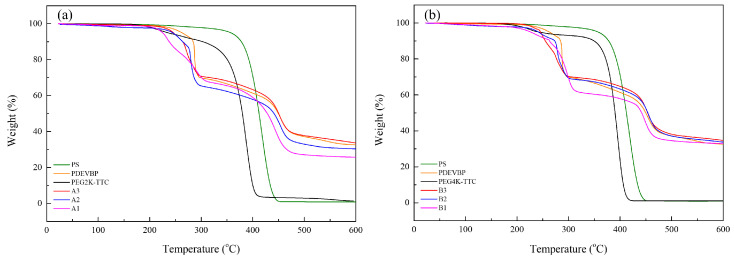
TGA measurements of (**a**) PEG2k-b-PDVBP, (**b**) PEG4k-b-PDVBP.

**Figure 11 polymers-12-01244-f011:**
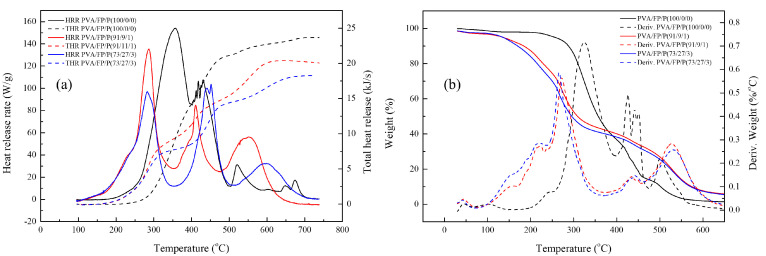
TGA measurements (**a**), Heat release rate (HRR) and total heat release (THR) (**b**) of PVA/PEG-b-PDEVBP; weight ratio of PVA/PEG-b-PDEVBP(FP, B2)/P (phosphorus contents) = PVA/FP/P.

**Table 1 polymers-12-01244-t001:** Polymerization and colloidal properties of polymer particles synthesized via the RAFT Emulsion homopolymerization of DEVBP.

Entry	Monomer wt. %	Mn (PEG)	[DEVBP]_0_/[PEG-TTC]_0_	P_theory_ %	Conversion ^a^ %	Molecular Weight			Size	Residual Weight at 500 °C (%) TGA analysis
						Theory ^b^	GPC (Mw/Mn)	^1^H NMR	D_z_ (nm) ^c^ (*Ð*) ^d^	
**A1**	**5**	**2K**	**32**	**9.0**	**89**	10,100	5200 (1.08)	9800	216 (0.22)	27.1
A2	9	2K	65	10.1	85	17,300	9400 (1.08)	16,800	1056 (0.68)	33
A3	16	2K	130	10.7	92	31,500	16,100 (1.07)	34,200	-	37.8
B1	5	4K	32	7.6	91	12,300	6700 (1.20)	12,200	194 (0.22)	34.5
B2	9	4K	65	9.1	82	18,800	11,000 (1.35)	18,100	302 (0.10)	37.1
B3	16	4K	130	10.2	87	35,000	14,100 (1.17)	33,200	576 (0.20)	38.1

Reaction condition: [PEG-TTC]_0_ = 5.5 mmol/L, [ACPA] = 1.65 mmol/L, temperature = 70 °C, time = 6h. ACPA as used initiator was neutralized by 3 mol equiv of NaHCO_3_. ^a^ Monomer conversion was determined by ^1^H NMR. ^b^ Theoretical number-average molar weight at the experimentally determined conversion; M.W_PEG-TTC_ + [DEVBP]_0_/[PEO-TTC]_0_
^c^ D_z_ is the average particle diameter from dynamic light scattering analysis. ^d^Polydispersity index (***Ð***) the particle size distribution.

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
