# Peer review of "Flame Retardant Submicron Particles via Surfactant-Free RAFT Emulsion Polymerization of Styrene Derivatives Containing Phosphorous"

_polymers, 2020, doi:10.3390/polym12061244_

Round 1

Reviewer 1 Report

In this manuscript, Lee, Paik and cowokers report the synthesis of funtional polystyrene nanoparticles by surfactant‐free RAFT emulsion polymerization and evaluate their flame retardant properties. Although the synthesis methodology is not really novel, the use of diethyl‐(4‐vinylbenzyl) phosphate as a hydrophobic monomer resulting excellent flame retardant property is interesting and therefore, is highly recommended for publishing on Polymers after revisions as noted.

  1. This work focus on surfactant‐free RAFT emulsion polymerization and the authors explain the disadvantages of surfactant RAFT emulsion polymerization very clear: “The final polymer product, however, has adverse effects on the properties of the products due to surfactant migration through the polymeric material, leading to degradation of their adhesive and mechanical properties. In addition, they also harm the environment.” After this, the authors should also discuss several advantages of surfactant‐free RAFT emulsion polymerization demonstrated recently including controlled size and shape for biomedical applications (Polymer Chemistry 2017, 1353-1363 and Small 2018, 1801702).
  2. In Figure 2, the authors should present panel a on the top (full wide) as it is very difficult to see the NMR peaks.
  3. In Figure 5, the NMR spectra should be labelled as Figure 4.
  4. In table 1, the theoretical molecular weight is much higher than the one determined by GPC while using polystyrene standard (some data are double). Please clarify this point.
  5. In Figure 6, the GPC traces show obvious tailing to the low molecular weight. Did the phosphate functional group affect the radical polymerization and lead to the tailing? A control experiment using styrene would be helpful to understand this data.
  6. In Figure 7, the baselines of GPC traces have been cut too early, which may affect the dispersity values.
  7. On page 7, the authors report an important conclusion that “particle size increased with increasing target molecular weight”. This finding has been reported and discussed in previous work, which should be cited here (Polymer Chemistry 2015, 3865-3874).
  8. Figure 9 shows the confocal images of PEG‐b‐P(DEVBP) nanoparticles using Nile red as a fluorescent agent. However, there is no experimental detail of how to load Nile red to these nanoparticles.
  9. In Figure 10, the use of the PEO-TTC as the control to compare the degradation is not appropriate as it does not consider the effect of polystyrene (PS). PEO-PS diblock (made in point 3) should be also useful here as the control for thermal degradation study.
  10. This work overall has been well-characterized except electron microscopy should be added to confirm the particle size and also show the particle shape.

Author Response

Dear Reviewer, 

We have added additional experimental such as TGA, micro cone calorimeter (MCC) for commercial PVA with PEG-b-PDEVBP nanoparticles., and we have corrected and updated manuscripts according to the editor and the reviewers.

Here is the response to your comment.

Reviewer #1.

In this manuscript, Lee, Paik and cowokers report the synthesis of funtional polystyrene nanoparticles by surfactant‐free RAFT emulsion polymerization and evaluate their flame retardant properties. Although the synthesis methodology is not really novel, the use of diethyl‐(4‐vinylbenzyl) phosphate as a hydrophobic monomer resulting excellent flame retardant property is interesting and therefore, is highly recommended for publishing on Polymers after revisions as noted.

Q1: This work focus on surfactant‐free RAFT emulsion polymerization and the authors explain the disadvantages of surfactant RAFT emulsion polymerization very clear: “The final polymer product, however, has adverse effects on the properties of the products due to surfactant migration through the polymeric material, leading to degradation of their adhesive and mechanical properties. In addition, they also harm the environment.” After this, the authors should also discuss several advantages of surfactant‐free RAFT emulsion polymerization demonstrated recently including controlled size and shape for biomedical applications (Polymer Chemistry 2017, 1353-1363 and Small 2018, 1801702).

A1: According to the reviewer’s comment, we have added description and citation referred the papers (Polymer Chemistry 2017, 1353-1363 and Small 2018, 1801702)(ref. 13, 14) as follows.

In line 59 of page 2.

Recently, surfactant-free RAFT emulsion polymerization was shown that the particle size and shape size can be controlled, and it was widely used in the biomedical field.[13,14]

Q2: In Figure 2, the authors should present panel a on the top (full wide) as it is very difficult to see the NMR peaks.

A2: According to the reviewer’s comment, we have corrected and rearranged for 1H and 31P NMR in Figure 2 as follow.

Q3: In Figure 5, the NMR spectra should be labelled as Figure 4.

A3: According to the reviewer’s comment, we have added peak label in Figure 5 as follows.

Q4: In table 1, the theoretical molecular weight is much higher than the one determined by GPC while using polystyrene standard (some data are double). Please clarify this point.

Q5: In Figure 6, the GPC traces show obvious tailing to the low molecular weight. Did the phosphate functional group affect the radical polymerization and lead to the tailing? A control experiment using styrene would be helpful to understand this data.

A4, 5: Thanks for reviewer’s comment. We have added and corrected for GPC analysis for PEG-b-PDEVBP as follows. We estimated that the theoretical molecular weight is much higher than the one in GPC trace because strong intermolecular interaction of the phosphate moiety of PEG-b-PDEVBP. The tailing in Figure 6 was also occurred by phosphate moiety.

In line 213 of page 8.

However, it was shown that the tailing to low molecular weight, and the molecular weight of all polymers (Figure 7, A1 ~ B3) was smaller in GPC than the actual molecular weight calculated by 1H NMR spectroscopy (Table 1). In previous paper, polymerization of styrene using PEG-TTC tailing was not observed at high monomer conversion and symmetrical curve was shown in GPC.[26] The hydrodynamic volume of PEG-b-PDEVBP series, a polymer of DEVBP derived from styrene, were very small compared with polystyrene standard. It was indicated that strong intermolecular interaction were occurred between the phosphate moieties of PEG-b-PDEVBP, leading to tailing toward low molecular weight (Figure 6).

Q6: In Figure 7, the baselines of GPC traces have been cut too early, which may affect the dispersity values.

A6: Thanks for reviewer’s comment. We have recalculated some time molecular weight and dispersity over wide range but the values of dipersity were same previous data.

Q7: On page 7, the authors report an important conclusion that “particle size increased with increasing target molecular weight”. This finding has been reported and discussed in previous work, which should be cited here (Polymer Chemistry 2015, 3865-3874).

A7: According to the reviewer’s comment, we have added content and citation of papers (Polymer Chemistry 2015, 3865-3874)(ref. 29) as follows.

In line 227 of page 8

Particle size increased with increasing molecular weight in emulsion RAFT polymerization previously,[29] the PDVBP particle size also increased with increasing target molecular weight (z- average size 216 ~ 1,056 nm).

Q8: Figure 9 shows the confocal images of PEG‐b‐P(DEVBP) nanoparticles using Nile red as a fluorescent agent. However, there is no experimental detail of how to load Nile red to these nanoparticles.

A8: According to the reviewer’s comment, we have added detailed experimental procedure to load Nile red in section 2.2 as follow.

In line 118 of page 4

To obtain the nanoparticles stained with Nile red, DEVBP containing 0.1wt% Nile red was used for RAFT emulsion polymerization.

Q9: In Figure 10, the use of the PEO-TTC as the control to compare the degradation is not appropriate as it does not consider the effect of polystyrene (PS). PEO-PS diblock (made in point 3) should be also useful here as the control for thermal degradation study.

A9: According to the reviewer’s comment, we have added TGA data for and discussion of comparison with PEG-b-PS from other reference as follows.

In line 239 of page 9

PEG-TTC 2k, 4k and polystyrene mainly degraded at 380oC and were degraded entirely at 400oC. The residue was 0 % above 400oC. Different thermal degradation behaviors were observed in PDVBP. The thermal degradation of PDVBP (P contents: 9 ~ 10.2%) started at 250oC to 300oC; the residue weight was above 60%. The initial degradation of PDVBP was contributed by the P-O-C structure functional group, P-O-C group of -PO4 is unstable than -PO1-3. Previously, -PO4 group was more effective for Phosphorus release and forming char [22]. The second thermal degradation occurred at 400 ~ 430 to 470oC, which was contributed by aromatic and PEG; the residue weights were 27.1 ~ 38.1 %(A1 ~ B3). Thermal degradation of aromatic group and PEG-TTC were occurred at high temperature (above 450oC) compared with polystyrene (410oC) and PEG-TTC (380oC). It was evidenced to form char by phosphorus contents and PEG-b-PDEVBP nanoparticles can provide flame retardant properties.

Q10: This work overall has been well-characterized except electron microscopy should be added to confirm the particle size and also show the particle shape.

Q10: Thanks for reviewer’s comment. We have attempted several time to obtain solid state nanoparticles by various method such as lyophilization, ultrasedimentation. But we didn’t obtain as solid state nanoparticles due to sticky and low Tg of PEG-b-PDEVBP

Sincerely, 

Taeyoon Kim

Reviewer 2 Report

Review of

Flame Retardant Nanoparticles via Surfactant‐free RAFT Emulsion Polymerization of Styrene Derivatives containing Phosphorous

by Taeyoon Kim, Joo‐Hyun Song, Jong‐Ho Back, Bongkuk Seo, Choong‐Sun Lim, Hyun‐Jong Paik and Wonjoo Lee.

Summary of the work

The topic of this manuscript fit well with the scope, presenting synthesis of new functional block copolymers using a controlled polymerization technique which might be of interest for flame retardant materials design.

More specifically, the authors describe the synthesis of diblock amphiphilic copolymers poly(ethylene glycol)-block- poly(diethyl‐(4‐vinylbenzyl) phosphate) (PEG-b-PDVEPB) using the reversible addition-fragmentation transfer (RAFT) polymerization induced-self-assembly (PISA) process in water. They specifically use a trithiocarbonate dodecyl-functionalized PEG macroRAFT agent to promote and control the RAFT-emulsion polymerization of the hydrophobic DEVBP monomer, presenting phosphorous moieties.

They perform polymerization, varying the DP of the 2nd block and the molar mass of the macroRAFT agent and present kinetics, evidencing a rather controlled polymerization. Then, they characterize the in situ formed nanoparticles using dynamic light scattering (DLS) and confocal microscopy and assess their behavior against temperature using thermogravimetry (TGA). It seems clear that presence of PDEVBP block with phosphore moieties induce an increased thermal resistance to high temperature, confirming their interest for flame retardant applications.

The manuscript is quite clear and the data and results are well documented.

Broad comments

The manuscript is overall well written but could be a little bit more structured to help the reader follow the logic of the results presentation.

For instance, the transition between description of data from literature (ref 21 to 24) and description of actual results is unclear.

The introduction could/should emphasize on the originality of this work or on the importance of the findings. At the moment, it does not stand out. (see comment/question n°3)

Concerning the presentation, the resolution of the graphs and the chemical structures presented could be enhanced for an easier comprehension.

In addition, to help the reader fully understand the objectives and the interest of this work, the authors could answer the questions listed below and add these informations in their publication.

General Questions

Q1: what is the purpose/advantage of forming a nanoparticle for these applications?

Q2 :  Why choosing to put the phosphorous block only in the core? Since it appears that the thermal resistance increase with the phosphorous content, why not synthesize a diblock copolymer presenting phosphorous units in both blocks?

Q3 : Compared to the literature, what is the advantage of this diblock copolymer? The authors should highlight the originality or interest of this work compared to the phosphorous-diblock copolymers already existing in the introduction

Q4 : From the conclusion, the authors state the obtention of “relatively stable particles”, did they asses the stability? If so how? And why is it important?

Specific comments

Scientifically, apart from some missing information in the experimental section (see questions/comments Q5 to Q10) the different studies are well documented and data seem reliable.

Questions section 2 Materials & Methods

Q5 : section 2.2 Can the authors mention the concentration of the solution analyzed in the particle size measurement (DLS)? Is it raw samples or diluted ones? Which methods was used to get the hydrodynamic diameter?

Q6: section 2.2 Can the authors specify if the TGA was performed on dried samples and if so, mention the process of drying at some point (in section 2.5?)?

Q7 : section 2.2 Additional informations concerning the GPC system could be of use (column informations for instance).

Q8 : section 2.3 what is the yield of such synthesis?

Q9 : section 2.5 The authors should add information regarding the monitoring of the polymerization such as "aliquots were withdrawn from the reaction media and analyzed by 1H NMR to monitor the monomer conversion”

Q10: Lastly, in Figure 6c, units should be added in the title of the x axis.

Author Response

Dear Reviewer, 

We have added additional experimental such as TGA, micro cone calorimeter (MCC) for commercial PVA with PEG-b-PDEVBP nanoparticles., and we have corrected and updated manuscripts according to the editor and the reviewers.

Here is the response to your comment.

Reviewer #2.

Summary of the work

The topic of this manuscript fit well with the scope, presenting synthesis of new functional block copolymers using a controlled polymerization technique which might be of interest for flame retardant materials design.

More specifically, the authors describe the synthesis of diblock amphiphilic copolymers poly(ethylene glycol)-block- poly(diethyl‐(4‐vinylbenzyl) phosphate) (PEG-b-PDVEPB) using the reversible addition-fragmentation transfer (RAFT) polymerization induced-self-assembly (PISA) process in water. They specifically use a trithiocarbonate dodecyl-functionalized PEG macroRAFT agent to promote and control the RAFT-emulsion polymerization of the hydrophobic DEVBP monomer, presenting phosphorous moieties.

They perform polymerization, varying the DP of the 2nd block and the molar mass of the macroRAFT agent and present kinetics, evidencing a rather controlled polymerization. Then, they characterize the in situ formed nanoparticles using dynamic light scattering (DLS) and confocal microscopy and assess their behavior against temperature using thermogravimetry (TGA). It seems clear that presence of PDEVBP block with phosphore moieties induce an increased thermal resistance to high temperature, confirming their interest for flame retardant applications.

The manuscript is quite clear and the data and results are well documented.

Broad comments

The manuscript is overall well written but could be a little bit more structured to help the reader follow the logic of the results presentation.

Q1: For instance, the transition between description of data from literature (ref 21 to 24) and description of actual results is unclear.

A1: According to the reviewer’s comment, we have corrected and added description for literature (ref 21 to 24) and description of actual results

Q2: The introduction could/should emphasize on the originality of this work or on the importance of the findings. At the moment, it does not stand out. (see comment/question n°3)

A2: According to the reviewer’s comment, we have corrected and added description of introduction the originality of this work. PO3 functionalized methacrylate have polymerized by RAFT polymerization so far, on the other hand we used PO4 fictionalized styrene and the phosphorus monomer was first used to RAFT emulsion polymerization.  

Q3: Concerning the presentation, the resolution of the graphs and the chemical structures presented could be enhanced for an easier comprehension.

A3: According to the reviewer’s comment, we have corrected the figures as a high resolution.

In addition, to help the reader fully understand the objectives and the interest of this work, the authors could answer the questions listed below and add these informations in their publication.

General Questions

Q4: what is the purpose/advantage of forming a nanoparticle for these applications?

A4: Thanks for reviewer’s comment. The PEG-b-PDEVBP nanoparticles could be used as additive for aqueous polymer solution, emulsion and dispersion to give flame retardant property. The nanoparticle is prepared by environmentally friend method(RAFT emulsion polymerization).  

Q5 : Why choosing to put the phosphorous block only in the core? Since it appears that the thermal resistance increase with the phosphorous content, why not synthesize a diblock copolymer presenting phosphorous units in both blocks?

A5: Thanks for reviewer’s comment. We have focused on phosphorus block as a hydrophobic and emulsion RAFT polymerization for styrene derivative phosphorus monomerdiblock copolymer presenting phosphorous units in both blocks and various morphologic block copolymers should be carried out in our next research.

Q6 : Compared to the literature, what is the advantage of this diblock copolymer? The authors should highlight the originality or interest of this work compared to the phosphorous-diblock copolymers already existing in the introduction

A6: Thanks for reviewer’s comment. Please see Answer #2 from Reviewer #2.

Q7 : From the conclusion, the authors state the obtention of “relatively stable particles”, did they asses the stability? If so how? And why is it important?

A7: Thanks for reviewer’s comment. We have corrected the conclusion as follows. In case of use PEG2K based macro-CTA, size dipersity was higher than PEG4K’s one in same DP or the sedimentation was shown at high DP (PEG2K, DP130)

Specific comments

Scientifically, apart from some missing information in the experimental section (see questions/comments Q5 to Q10) the different studies are well documented and data seem reliable.

Sincerely, 

Taeyoon Kim

Questions section 2 Materials & Methods

Q8 : section 2.2 Can the authors mention the concentration of the solution analyzed in the particle size measurement (DLS)? Is it raw samples or diluted ones? Which methods was used to get the hydrodynamic diameter?

A8: Thanks for reviewer’s comment, we have added detail measurement process. All samples were measured after dilute 3 mg (solid contents)/ml in water and were analyzed by DLS in size measurements mode using a disposal cuvette.

Q9: section 2.2 Can the authors specify if the TGA was performed on dried samples and if so, mention the process of drying at some point (in section 2.5?)?

A9: According to the reviewer’s comment, the samples were dried by the lyophilization and we have added this process in section 2.2 as follows

In section 2.2

2.2. Characterization

1H NMR and 31P NMR spectra were recorded in CDCl3 on Bruker 300 Mhz spectrometer. FT-IR spectra were obtained at a resolution of 4 cm-1 with ThermoFisher Nicolet 6700 spectrometer in the wavenumber rage of 4000 ~ 400 cm-1. The powder samples were incorporated into KBr pellets and IR measurements were performed. Thermal degradation was examined using a TA Q500 thermogravimetric analyzer (TGA) with heating rate of 10oC/min under nitrogen atmosphere and samples were dried by the lyophilization .The heat resistance was studied with pyrolysis combustion flow calorimetry (PCFC, Fire Testing Technology Limited, West Sussex, UK) with the ASTM D7309 method. The samples were prepared by the mixing 10 wt% PVA solution with PEG-b-PDEVBP emulsion followed by the lyophilization of the mixture and then loaded in the instrument is heated at a rate of 1 C/s from 100 to 750oC to obtain heat release rate (HRR) and total heat release (THR). Molecular weight and dispersity were determined by (GPC), conducted with a Agilent 1260 isopump and Agilent 1260 differential refractometer using Agilent columns (2 × PLgel 5 μm MIXED-D, 7.5 × 300 mm) in THF as an eluent at 40°C and at a flow rate of 1 mL/min. Linear polystyrene standards were used for calibration. Particles size measurement was performed on Malvern Nano-ZS90. Multi-photon confocal microscopy images were acquired using a ZEISS LSM 780 configuration 16 NLO microscope with Nile red as fluorescent agent. To obtain the nanoparticles stained with Nile red, DEVBP containing 0.1wt% Nile red was used for RAFT emulsion polymerization.

Q10 : section 2.2 Additional informations concerning the GPC system could be of use (column informations for instance).

A10: Thanks for reviewer’s comment, Thanks for reviewer’s comment, we have added mesuremetns condition of GPC analysis. . Please see Answer #9 from Reviewer #2. 

Q11 : section 2.3 what is the yield of such synthesis?

A11: Thanks for reviewer’s comment, yield of block copolymer samples were nearly 90%.

Q12: section 2.5 The authors should add information regarding the monitoring of the polymerization such as "aliquots were withdrawn from the reaction media and analyzed by 1H NMR to monitor the monomer conversion”

A12: Thanks for reviewer’s comment, Thanks for reviewer’s comment, we have added information for detail method for polymerization conversion by 1H NMR in section 2.6 as follows.

2.6. RAFT emulsion polymerization of diethyl 4-vinylphenyl phosphate (DEVBP)

PEG-TTC 4k (0.27 g, 0.060 mmol), ACPA (0.012 g, 0.042 mmol) were added to a 100 mL schlenk flask and dissolved in water. DEVBP (1.06 g, 3.94 mmol) was added to the reaction mixture and dispersed. An aqueous stock solution of neutralized ACPA by NaHCO3 (3.5 molar equivalents with respect to ACPA) was prepared and added. Freeze-pump-thaw cycles were performed 3 times to remove oxygen. After the flask was backfilled with nitrogen, the polymerization was carried out at 70oC for 6 h. Adequate samples were periodically withdrawn from the reactor analyzed by by 1H NMR to evaluate polymerization kinetics, conversion was calculated by the comparsion vinyl peak with methylene peaks of DEVBP.

Q13: Lastly, in Figure 6c, units should be added in the title of the x axis.

A:13 Thanks for reviewer’s comment, We have corrected and added title of the x axis in Figure 6c as follows.

Round 2

Reviewer 1 Report

Thanks to the authors' efforts on revising the manuscript, the manuscript has been significantly improved and now is recommended to be published in Polymers.